# Transfer Learning using Spectral Convolutional Autoencoders on Semi-Regular Surface Meshes

**Sara Hahner**
Fraunhofer Center for Machine Learning and SCAI, Sankt Augustin
`sara.hahner@scai.fraunhofer.de`

**Felix Kerkhoff**
Johannes Kepler Universität, Linz

**Jochen Garcke**
Fraunhofer Center for Machine Learning and SCAI, Sankt Augustin
Institut für Numerische Simulation, Universität Bonn

## Abstract

The underlying dynamics and patterns of 3D surface meshes deforming over time can be discovered by unsupervised learning, especially autoencoders, which calculate low-dimensional embeddings of the surfaces. To study the deformation patterns of unseen shapes by transfer learning, we want to train an autoencoder that can analyze new surface meshes without training a new network. Here, most state-of-the-art autoencoders cannot handle meshes of different connectivity and therefore have limited to no generalization capacities to new meshes. Also, reconstruction errors strongly increase in comparison to the errors for the training shapes. To address this, we propose a novel spectral CoSMA (Convolutional Semi-Regular Mesh Autoencoder) network. This patch-based approach is combined with a surface-aware training. It reconstructs surfaces not presented during training and generalizes the deformation behavior of the surfaces' patches. The novel approach reconstructs unseen meshes from different datasets in superior quality compared to state-of-the-art autoencoders that have been trained on these shapes. Our transfer learning errors on unseen shapes are 40% lower than those from models learned directly on the data. Furthermore, baseline autoencoders detect deformation patterns of unseen mesh sequences only for the whole shape. In contrast, due to the employed regional patches and stable reconstruction quality, we can localize where on the surfaces these deformation patterns manifest.

## 1 Introduction

We study the deformation of surfaces in 3D, which discretize human bodies, animals, or work pieces from computer aided engineering. Using autoencoders as a method for unsupervised learning, we analyze and detect patterns in the deformation behavior by calculating low-dimensional features. Since surface deformation is locally described by the same physical rules, we want to study the deformation patterns of unseen shapes by transfer learning. In our context, the broad term transfer learning means that an autoencoder should be able to analyze new surface meshes without being trained again.

While two-dimensional surfaces embedded in $\mathbb{R}^3$ are locally homeomorphic to the two-dimensional space, they are of non-Euclidean nature. Their representation by surface meshes lacks the regularity of pixels describing images, which is so convenient for 2D CNNs [1]. This is why existing methods for unsupervised learning for irregularly meshed surface meshes depend on the mesh connectivity when defining pooling or convolutional operators. For this reason, a trained mesh autoencoder cannot be applied to a surface that is represented by a different mesh, although the local deformation behavior might be similar.

S. Hahner, F. Kerkhoff, and J. Garcke, Transfer Learning using Spectral Convolutional Autoencoders on Semi-Regular Surface Meshes. *Proceedings of the First Learning on Graphs Conference (LoG 2022)*, PMLR 198, Virtual Event, December 9–12, 2022.

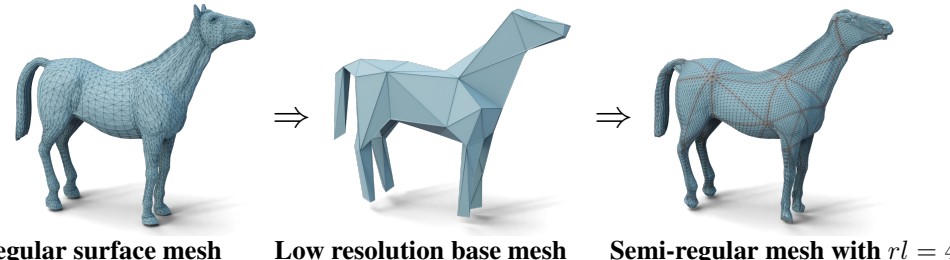

**Irregular surface mesh**   **Low resolution base mesh**   **Semi-regular mesh with** $rl = 4$

**Figure 1:** Remeshing of the horse template mesh. In the semi-regular mesh, the boundaries of the regularly meshed patches are highlighted in gray.

In [2] we introduced a mesh autoencoder for semi-regular meshes of different sizes. The semi-regular surface representations enforce some local mesh regularity and are made up of regularly meshed patches as illustrated in Figure 1, which allows the application of a patch-wise approach. However, the reconstruction quality decreases by a factor of 4 when applying this mesh autoencoder to new meshes and shapes that have not been used during training. This limits the method's application for unseen shapes.

Additionally, baseline mesh autoencoders for deforming shapes do not provide an understanding or explanation about which surface areas lead to the patterns in the embedding space. The embeddings represent the entire shape. Nevertheless, when identifying and analyzing deformation patterns, it is of particular relevance where on the surfaces these patterns manifest.

Our current work remedies these gaps by adopting the patch-based framework for semi-regular meshes and choosing a spectral graph convolutional filter [3] projecting vertex features to the Laplacian eigenvector basis in combination with a surface-aware training. Since the spectral filters consider the entire patch, the network generalizes better in comparison to a spatial approach, whose filters consider smaller $n$-ring neighborhoods. This improves the quality and smoothness of the reconstruction results when being applied to unknown meshes and the errors are 40% lower than errors from models learned directly on the data. Although spectral graph neural network methods require fixed mesh connectivity, the patch-based and therefore mesh-independent approach is not limited by this constraint. This is because the filters are applied to the regular substructures of semi-regular mesh representations of the surfaces as in [2]. Furthermore, our patch-based approach allows us to correlate patch-wise embeddings with the embedding of the entire shape (Figure 2). This way we localize and understand where on the surfaces the deformation patterns manifest, these are visible in the low-dimensional representation.

The research objectives can be summarized as a) the definition of a spectral convolutional autoencoder for semi-regular meshes (spectral CoSMA) and a surface-aware training loss, by this means b) improving the transfer learning, generalization capability and runtime of baseline mesh autoencoders, and c) localizing the deformation patterns visible in the low-dimensional embedding on the surfaces.[1]

Further on in section 2, we discuss work related to learning features from meshed geometry. Additionally, we present relevant characteristics of surface meshes for CNNs and the semi-regular remeshing, In section 3 we present the definition of our spectral CoSMA and the surface-aware loss calculation. Results for different datasets containing meshes with different connectivity are presented in section 4.

## 2 Related Work: Handling Surface Meshes by Neural Networks

Surfaces are generally represented either in form of point clouds or by a surface mesh, which is defined by faces connecting vertices to each other. We consider the representation via meshes because their faces describe the underlying surface [4, 5].

---

[1]Source code and used data available at: `https://github.com/Fraunhofer-SCAI/spectral_CoSMA`

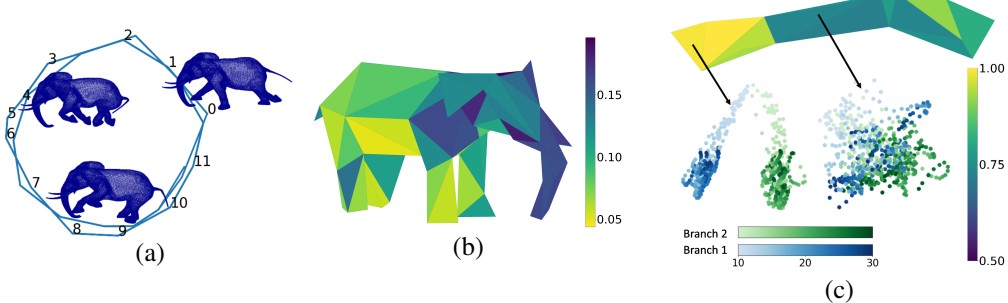

**Figure 2:** (a) 2D Embedding of the low-dimensional representation of the whole elephant over time. (b) Highlighting the distance of the patch-wise embeddings to the embedding of the whole shape. (c) Patch-wise score for the TRUCK's front beam from Figure 5 at $t = 24$. Only the patch with the high score manifests the deformation in two patterns. This is visible in the example patches with high and low scores. The embedding's colors encode timestep and branch.

## 2.1 Convolutional Networks for Surfaces

Surface meshes can be viewed as graphs, and hence graph-based convolutional methods are often applied to meshes. Generally, convolutional networks for graphs can be separated into spectral and spatial ones, of which [1, 6, 7] give an overview. Spatial convolutional methods for graphs aggregate features based on a node's spatial relations, which allows generalization across different mesh connectivities [7, 8]. Spectral approaches, on the other hand, interpret information on the vertices as a signal propagating along the vertices. They exploit the connection of the graph Laplacian and the Fourier basis and vertex features are projected to the Laplacian eigenvector basis, where filters are applied [9]. Instead of explicitly computing Laplacian eigenvectors, the authors of [3] use truncated Chebyshev polynomials, while in [10] only first-order Chebyshev polynomials are used. These spectral methods require fixed connectivity of the graph. If not, the adjacency matrix and consequently the Laplacian eigenvector basis change. Furthermore, there are network architectures only for surface meshes, e.g. DiffusionNet [11] and HodgeNet [12], which are applied for classification, mesh segmentation, and shape correspondence. Nevertheless, these architectures cannot be implemented directly into autoencoders, because of missing mesh pooling operators.

## 2.2 Neural Networks for Semi-Regular Surface Meshes

Semi-regular triangular surface meshes, also known as meshes with subdivision connectivity, come with a regular local structure and a hierarchical multi-resolution structure. In section 2.4, we provide a more detailed definition. The Spatial CoSMA [2] and SubdivNet [13] take advantage of the local regularity of the patches by defining efficient mesh-independent pooling operators and using 2D convolution. By inputting the patches separately into the network, we define in [2] an autoencoder pipeline that is independent of the mesh size. [13] apply self-parametrization using the MAPS algorithm [14] to remesh watertight manifold meshes without boundaries. On the other hand, we apply in [2] a remeshing algorithm that works for meshes with boundaries and coarser base meshes.

## 2.3 Mesh Convolutional Autoencoders

Some of the first convolutional mesh autoencoders have been introduced in [15] and [16] (CoMA). The authors of CoMA introduced mesh downsampling and mesh upsampling layers for pooling and unpooling, which are combined with spectral convolutional filters using truncated Chebyshev polynomials as in [3]. The Neural3DMM network presented in [4] improves those results using spiral convolutional layers. By manually choosing latent vertices for the embedding space, [17] define an autoencoder that allows interpolating in the latent space. All the above-mentioned mesh convolutional autoencoders work only for meshes of the same size and connectivity because the pooling and/or convolutional layers depend on the adjacency matrix. We showed in [2] that the latter methods are not able to learn data with greater global variations in comparison to our patch-based approach, which generalizes and reconstructs the deformed meshes to superior quality. Additionally, our architecture can be applied to unseen meshes of different sizes. The MeshCNN architecture [5]

can be implemented as an encoder and decoder. Nevertheless, the pooling is feature dependent and therefore the embeddings can be of different significance. [18] and [19] achieve particularly good results in shape reconstruction and completion by representing shapes using signed distance functions and other implicit representations. As these approaches are representing whole shapes using a single fixed-length vector, their generalization and scalability are often limited, which is why our work is mainly focused on mesh-based methods.

### 2.4 Definition of Semi-Regular Meshes

The irregularity of surface meshes gives rise to difficulties when handling them with a neural network. Whereas CNNs in 2D [20, 21] apply the same local filters to local neighborhoods of selected pixels of the image and shift them horizontally and vertically, this is not applicable to surface meshes [22]. In comparison to 2D images, surface meshes lack global regularities, because they are not defined along a global grid, and local neighborhoods can have any size and arrangement as long as they are locally Euclidean.

One cannot enforce a regular mesh discretization for every surface in $\mathbb{R}^3$, which would lead to an underlying global grid [23]. This is why [2, 24] proposed to enforce a similar structure in the local neighborhoods by choosing a semi-regular representation of the surface. In this way, an efficient application of convolution on surface meshes becomes possible. Note that remeshing the polygonal mesh only changes the representation of the objects, allowing just small, bounded distortions. The considered surface embedded in $\mathbb{R}^3$ is the same, but now represented by a different discrete approximation.

Following the definition in [25], we call a surface mesh semi-regular if we can convert it to a low-resolution mesh by iteratively merging four triangular faces into one. Consequently, all vertices of the semi-regular mesh except for the ones remaining in the low-resolution mesh are regular (i.e. have six neighbors). Vice versa, the regular subdivision of a possibly irregular low-resolution mesh yields a semi-regular mesh. Such a regular subdivision can be achieved by inserting a vertex on each edge and splitting each original triangle face into 4 sub-triangles. [13, 26] refer to this property as Loop subdivision connectivity of the semi-regular mesh. The subdivision connectivity makes semi-regular meshes particularly useful for multiresolution analysis and directly implies a suitable local pooling operator on semi-regular meshes (see section 3.2).

### 2.5 Semi-Regular Remeshing

There are different remeshing algorithms, for example, Neural Subdivision [24] or MAPS [14]. Nevertheless, we cannot apply these, because they only work for closed surfaces without boundaries and fail for base meshes as coarse as ours. Therefore, we apply the remeshing from [2]. The algorithm iteratively subdivides a coarse approximation of the original irregular mesh (see Figure 1). The resulting semi-regular mesh is fitted to the original mesh using gradient descent on a loss function based on the chamfer distance. The refinement level $rl$ states the number of times each face of the coarse base mesh is iteratively subdivided. The number of faces in the final semi-regular mesh is $n_F^{semireg} = 4^{rl} * n_F^c$, with $n_F^c$ being the number of faces describing the coarse base mesh. We choose the refinement level $rl = 4$, which leads to finer meshes compared to [2], where we used $rl = 3$.

After the remeshing, all vertices that are newly created during the subdivision have six neighbors. Therefore, the resulting mesh is semi-regular or has subdivision connectivity.

## 3 Spectral CoSMA

The network handles the regional patches separately, which allows us to handle meshes of different sizes, as in [2]. We describe how the graph convolution is combined with the padding and the pooling of the patches. The building blocks are set together to define the spectral CoSMA (Spectral Convolutional Semi-Regular Mesh Autoencoder). Also, we introduce our surface-aware training loss to consider the patch-wise reconstructions as part of the entire mesh.

### 3.1 Spectral Chebyshev Convolutional Filters

We apply fast Chebyshev filters [3], as in [16], with the distinction that we are using them to perform spectral convolutions on the regional patches instead of the entire mesh. The approach in [3] performs

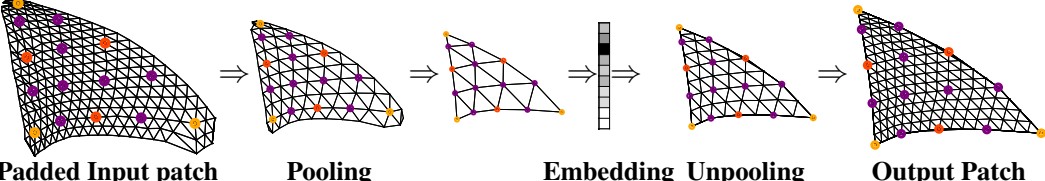

**Figure 3:** Resolution of the regularly meshed patches inside the spectral CoSMA. The encoder pools the patches twice by undoing subdivision. In the decoder, the unpooling increases the resolution again by subdivision. The orange vertices are the vertices from the irregular base mesh. Red and purple vertices have been created during the 1$^{st}$ and 2$^{nd}$ refinement steps.

spectral decomposition using spectral filters and applies convolutions directly in the frequency space. The spectral filters are approximated by truncated Chebyshev polynomials, which avoids explicitly computing the Laplacian eigenvectors and, by this means, reduces the computational complexity.

We justify this different convolution on the patches, compared to [2], by the intuition that spectral filters encode information of a whole patch and the general characteristics of its deformations, whereas in comparison spatial convolution considers just the local neighborhood around a vertex. Additionally, this spectral approach uses only the first few Chebyshev polynomials of the lowest degree, that resemble the lowest frequencies [27]. This is convenient when reconstructing surfaces, especially densely meshed ones, which tend to be relatively smooth in the local neighborhoods and have few features of high frequency.

The decomposition using spectral filters is dependent on the adjacency matrix, which restricts the transfer learning of learned spectral graph convolution to meshes of the same connectivity. Nevertheless, the adjacency matrix of the patches of our semi-regular meshes is always the same for one refinement level. This allows us to train the filters for all patches together and to apply them to unseen meshes.

### 3.2 Pooling and Padding of the Regular Patches

We apply the patch-wise average pooling and unpooling from [2] that takes advantage of the multi-scale structure of the semi-regular meshes. The subdivision connectivity guarantees that every 4 faces can be uniformly pooled to 1. The remaining vertices take the average of their own value and the values of the neighboring vertices that are removed. The unpooling operator subdivides the faces and the newly created vertices are assigned the average value of neighboring vertices from the lower-resolution mesh patch, see Figure 3. A similar pooling and unpooling operator is also applied by [13], where the information is saved on the faces.

The padding is crucial for the network to consider the regional patches in a larger context. Since the network handles the patches separately, we consider the features of the neighboring patches in a padding of size 2 as in [2]. If the vertices are boundary vertices, we decide to pad the patch with the boundary vertices' features.

### 3.3 Network Architecture

While using specialized pooling and convolution techniques for the regular patches, the general structure of our network architecture is inspired by [2, 16]. Our autoencoder architecture combines spectral Chebyshev convolutional filters with the described pooling technique to process the padded regular patches of a semi-regular mesh. The autoencoder compresses every padded patch, which corresponds to one face of the low-resolution mesh, from $\mathbb{R}^{276 \times 3}$ ($rl = 4$) to an $hr = 10$ dimensional latent vector and reconstructs the original padded patch from the latent vector.

The encoder consists of two blocks containing a Chebyshev convolutional layer followed by an average pooling layer and an exponential linear unit (ELU) as an activation function [28]. The output of the second encoding block is mapped to the latent space by a fully connected layer.

The decoder mirrors the structure of the encoder by first applying a fully connected layer, which transforms the latent space vector back to a regular triangle representation with refinement level

**Table 1:** Structure of the autoencoder for refinement level $rl = 4$, number of Chebyshev polynomials $K = 6$ and hidden representation of size $hr = 10$. The bullets • reference the corresponding batch size. The data's last dimension is the number of vertices considered for each padded patch.

| Encoder Layer | Output Shape | Param. | Decoder Layer | Output Shape | Param. |
|---|---|---|---|---|---|
| Input | $(\bullet, \ 3, 267)$ | 0 | Fully Connected | $(\bullet, 2^5, \ 15)$ | 5280 |
| ChebConv | $(\bullet, 2^4, 267)$ | 304 | Unpooling | $(\bullet, 2^5, \ 78)$ | 0 |
| Pooling | $(\bullet, 2^4, \ 78)$ | 0 | ChebConv | $(\bullet, 2^5, \ 78)$ | 6176 |
| ChebConv | $(\bullet, 2^5, \ 78)$ | 3104 | Unpooling | $(\bullet, 2^5, 267)$ | 0 |
| Pooling | $(\bullet, 2^5, \ 15)$ | 0 | ChebConv | $(\bullet, 2^4, 267)$ | 3088 |
| Fully Connected | $(\bullet, 10)$ | 4810 | ChebConv | $(\bullet, \ 3, 267)$ | 291 |

$rl = 2$. Afterward, two decoding blocks consisting of an unpooling layer followed by a convolutional layer transform the coarse triangle representation back to the original padded patch representation. Finally, another Chebyshev convolutional layer is applied without activation function to reconstruct the original patch coordinates by reducing the number of features to three dimensions.

All Chebyshev convolutional layers use $K = 6$ Chebyshev polynomials. Table 1 gives a detailed view of the structure of the network together with the parameter numbers per layer, which sum up to 23,053. Figure 3 illustrates the patch sizes inside the autoencoder. Note that we are able to handle non-manifold edges of the coarse base mesh because the patches, whose interiors by construction have only manifold-edges, are fed separately.

This spectral CoSMA architecture can handle all surface meshes that have been remeshed into a semi-regular mesh representation of the same refinement level. By handling the regional padded patches separately, this workflow is independent of the original irregular mesh connectivity thanks to the remeshing and patch-wise handling.

### 3.4  Surface-Aware Loss Calculation

Note, in [2] we employed for the patch-based spatial CoSMA a patch-wise mean squared error as the training loss. But, that loss calculation is not keeping track of multiple appearances of the vertices in the patch boundaries, whose errors are weighted higher than in the interior of the patches. Therefore, it is not surface-aware and not considering the patches as part of the entire mesh but separately. By weighting the vertex-wise error in the training loss with one divided by the vertices' number of appearances in the different patches, we employ a surface-aware error for training, whose definition is provided in section A.2. This reduces the P2S error by avoiding artifacts and errors due to the overemphasis of the patch boundaries, as visible in the ablation study and Figure 8. Note that only due to the improved reconstruction quality of the spectral approach one notices these artifacts.

## 4  Experiments

We test our spectral CoSMA for semi-regular meshes using an experiment setup similar to [2] on four different datasets and compare our reconstruction errors to state-of-the-art surface mesh autoencoders.

### 4.1  Datasets

**GALLOP:** The dataset contains triangular meshes representing a motion sequence with 48 timesteps from a galloping horse, elephant, and camel [29]. The galloping movement is similar but the meshes representing the surfaces of the three animals are different in connectivity and the number of vertices. This is why the baseline autoencoders have to be trained three times. The surface approximations are remeshed to semi-regular meshes with refinement level $rl = 4$ for each animal. The new meshes are still of different connectivity, but all are made up of regional regular patches. Table 9 lists the resulting numbers of vertices. We normalize the semi-regular meshes to $[-1, 1]$ as in [2]. Before inputting the data to the CoSMAs, every patch is translated to zero mean. We use the first 70% of the galloping sequence of the horse and camel for training. The architecture is tested on the remaining 30% and the whole sequence of the elephant, which is never seen during the training for the CoSMAs.

**Table 2:** Point to surface (P2S) errors ($\times 10^{-2}$) between reconstructed unseen semi-regular meshes ($rl = 4$) and original irregular mesh and their standard deviations for three different training runs. [4, 13, 16] have to be trained per mesh; we and [2] train one network for all three animals in the GALLOP dataset. $^*$: the elephant has not been seen by the network during training.

| Mesh Class | CoMA [16] | Neural3DMM [4] | SubdivNet [13] | Spatial CoSMA [2] | Ours |
|---|---|---|---|---|---|
| FAUST | $0.7073 + 1.751$ | $0.4064 + 0.921$ | $2.8190 + 4.699$ | $0.0224 + 0.045$ | $\mathbf{0.0031} + 0.006$ |
| Horse | $0.0053 + 0.017$ | $0.0096 + 0.045$ | $0.0113 + 0.025$ | $0.0078 + 0.012$ | $\mathbf{0.0022} + 0.005$ |
| Camel | $0.0075 + 0.023$ | $0.0145 + 0.056$ | $0.0113 + 0.024$ | $0.0091 + 0.014$ | $\mathbf{0.0030} + 0.006$ |
| Elephant | $0.0101 + 0.031$ | $0.0147 + 0.057$ | $0.0145 + 0.032$ | $0.0316 + 0.068^*$ | $\mathbf{0.0054} + 0.012^*$ |

**Table 3:** P2S errors ($\times 10^{-2}$) for three different training runs. Additionally, the Euclidean P2S error (in cm) is given. $^*$: the entire YARIS dataset has not been seen by the network during training.

| Dataset | Component Lengths | Spatial CoSMA [2] | | Ours | |
|---|---|---|---|---|---|
| | | Test P2S | Eucl. E. | Test P2S | Eucl. E. |
| TRUCK | 135–370 cm | $0.0660 + 0.117$ | 2.76 cm | $0.0013 + 0.003$ | 0.26 cm |
| YARIS$^*$ | 21–91 cm | $0.2061 + 0.438$ | 0.84 cm | $0.0375 + 0.088$ | 0.31 cm |

**FAUST:** The dataset contains 100 meshes [30], which are in correspondence to each other. The irregular surface meshes represent 10 different bodies in 10 different poses. For the experiments, we consider two unknown poses of all bodies (20% of the data) in the testing set. The meshes are remeshed and normalized in the same way as for the GALLOP dataset.

**TRUCK and YARIS:** In a car crash simulation the car components, which are generally represented by surface meshes, often deform in different patterns. Every component is discretized by a surface mesh, while the local deformation is described by the same physical rules. Following [2], the TRUCK dataset contains 32 completed frontal crash simulations and 6 components, the YARIS dataset contains 10 simulations and 10 components [31]. 30 simulations and 70% of the timesteps of the TRUCK dataset are included in the training set. The remaining samples from the TRUCK dataset and the entire YARIS dataset, representing a different car, are considered for testing. For this setup, the authors of [2, 32] detect patterns in the deformation of the TRUCK and YARIS components. We normalize the meshes that discretize car components to zero mean and range $[-1, 1]$ relative to the coordinates' ratio. Every patch is translated to zero mean.

### 4.2 Training Details

We train the network (implemented in Pytorch [33] and Pytorch Geometric [34]) with the adaptive learning rate optimization algorithm [35]. For the GALLOP and the FAUST dataset, we use a learning rate of 0.0001 and train for 150 epochs using a batch size of 100. For the TRUCK data, we choose a batch size of 100 combined with a learning rate of 0.001 for 300 epochs, since the variation inside the dataset is higher. We minimize the surface-aware loss between the original and reconstructed regional patches of the surface mesh without considering the padding. To augment the data in the case of the GALLOP and the FAUST dataset, we rotate the regional patches by $0°$, $120°$, and $240°$.

Our architecture requires at least 50% fewer parameters than the CoMA, Neural3DMM, and SubdivNet networks, because for increasing $rl$ and consequently finer meshes, the CoSMAs require only a few parameters more in the linear layers (compare Tables 9 and 10 in the supplementary material). This is because the patches and convolutional filters share the parameters. The spectral CoSMA approach requires 15% fewer parameters than the spatial CoSMA approach. The runtime analysis and ablation study justifying parameter choices are provided in section A.1.

### 4.3 Reconstructions of the Meshes

The mean squared error between true and reconstructed vertices of the semi-regular mesh allows a comparison of different methods only if the same remeshing result is used. In difference to [2],

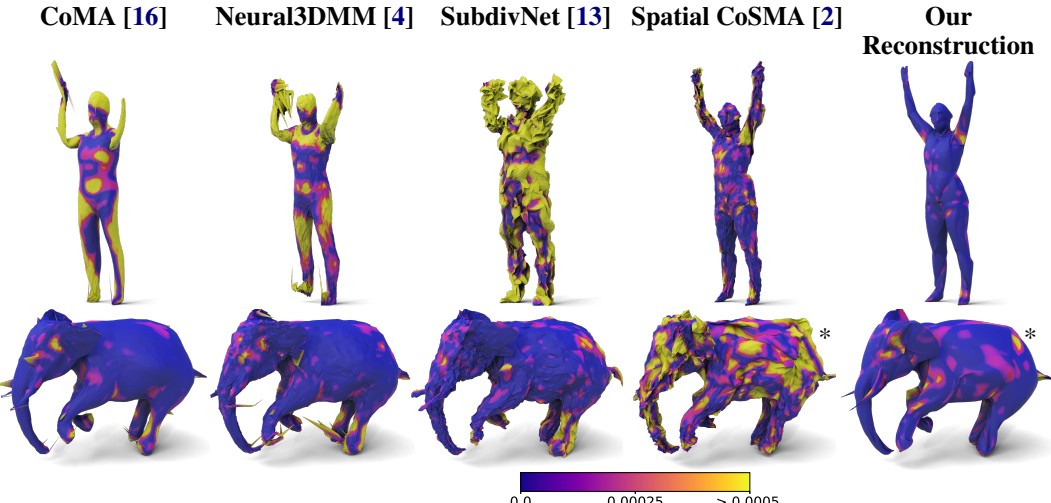

**Figure 4:** Reconstructed unknown FAUST pose and elephant test sample at $t = 43$ by CoMA, Neural3DMM, SubdivNet Autoencoder, spatial CoSMA, and our network. P2S error of the reconstructed faces is highlighted. More reconstruction examples are given in the supplementary material.
*: The elephant's mesh has not been presented during training to spatial CoSMA and our network.

we compare the reconstructed semi-regular mesh directly to the original irregular surface mesh by calculating a point to surface error (P2S). We average the mean squared errors between the vertices of the semi-regular mesh and their orthogonal projections to the surface described by the irregular mesh. This allows us to compare the reconstruction errors when using different remeshings or refinements.

Besides CoMA [16] and Neural3DMM [4], we use an additional baseline semi-regular mesh autoencoder using our network's architectures with the pooling and convolutional layers from SubdivNet [13] to process the entire meshes, see Table 12. In Table 2 we compare the autoencoders for the GALLOP and FAUST datasets in terms of the P2S errors of reconstructed test samples, whose 3D coordinates lie in the range $[-1, 1]$. Our network reduces the test reconstruction error for the GALLOP and FAUST datasets by more than 50% and 80% respectively, if the shape is presented to the autoencoder during the training. For unknown poses from the FAUST dataset, the limbs' positions are reconstructed inaccurately by the CoMA, Neural3DMM, and SubdivNet autoencoders. Especially if the pose is not similar to training poses, their reconstruction fails, as Figures 4 and 9 illustrate.

The spectral CoSMA's reconstructions are generally smoother than the ones from the spatial CoSMA, which reduces the reconstruction errors. Figure 10 in the supplementary material shows that the reconstructed patch using spectral filters, which encode the connectivity of the whole patch in the Chebyshev polynomials, is smoother than the spatial reconstruction, where the convolutional kernels only consider the close neighborhood. Because the spatial CoSMA uses $hr = 8$ and no surface-aware loss, we also list our reconstruction errors using these parameters in the ablation study for comparison.

**Transfer Learning to Other Meshes:** Our spectral CoSMA and the spatial CoSMA are the only networks that can reconstruct an unseen shape of different connectivity. The elephant's mesh has never been presented to our network. Nevertheless, the reconstruction error is almost as low as when training the spectral CoSMA only on the elephant, see Table 5 in section A.1. Even though trained on the elephant, the baselines' reconstructions are worse and unstable in the legs, as Figure 4 illustrates. The spatial CoSMA's reconstructions of the unseen elephant are inferior to all the other networks, although the reconstructions of the known camel and horse are of similar quality to the other baselines. This highlights the improved transfer learning capability of the spectral approach.

Since the patch-wise deformations of GALLOP and FAUST are both of natural origin, we test the out-of-distribution generalization of our spectral CoSMA. We train it on one and attempt reconstruction on the other dataset. The results are discussed in the supplementary section A.3.

Since the TRUCK and YARIS datasets contain 16 different meshes, the reconstruction results are only compared between the CoSMA architectures. Table 3 lists the average P2S errors for the TRUCK

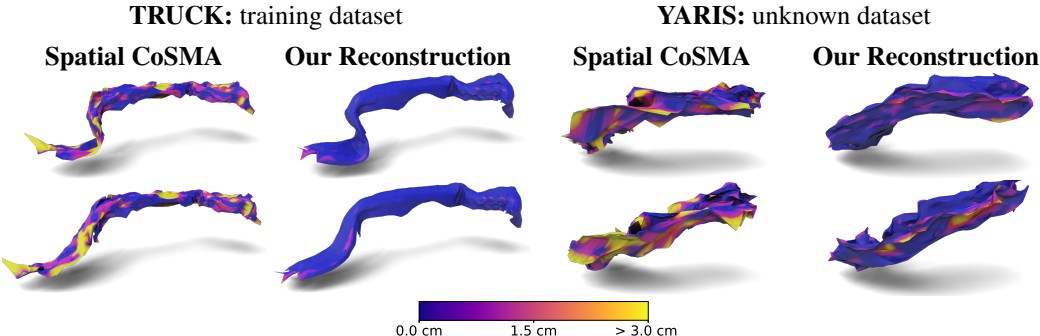

**Figure 5:** Reconstructed front beams from the TRUCK (length of 150 cm) at time $t = 24$ (test sample) from two crash simulations representing different deformation behavior and from the YARIS (length of 65 cm) at $t = 15$. The average Euclidean P2S error (in cm) of the faces is highlighted.

and YARIS datasets between the components scaled to range $[-1, 1]$ and in cm. The entire YARIS dataset has never been presented to the network during training. The results on the YARIS in Figure 5 also show that our network not only reconstructs smoother surfaces in comparison to the spatial CoSMA but also has higher generalization capacities. A comparison of the results for refinement levels $rl = 3$ and $rl = 4$ for the TRUCK and YARIS datasets (see Table 11 in the supplementary section A.5) shows the stability of the results from our spectral CoSMA. For the spatial CoSMA on the other hand, the reconstruction quality decreases when increasing the refinement level. This is due to the fixed kernel size of 2. Since the mesh is finer, the considered neighborhoods by a spatial filter using kernel size 2 cover smaller areas of the surface. The spectral CoSMA considers the entire patches in spectral representation. Therefore, an increase in the refinement level does not impair the reconstruction quality.

## 4.4 Low-dimensional Embedding

We project the patch-wise hidden representations of size $hr$ into the two-dimensional space using the linear dimensionality reduction method Principal Component Analysis (PCA) [36]. Then we compare these patch-wise results to the 2D embedding of the whole shape over time, by concatenating the hidden patch-wise representations and then applying PCA.

The time-dependent embedding for the unseen elephant from the GALLOP dataset exhibits a periodic galloping sequence, visualized in Figure 2 (a). We compare how similar the 2D patch-wise embeddings are to the 2D embedding for the entire shape, to determine how important the deformation of the patch is for the general deformation behavior of the whole shape. The patch-wise distance (low distance in yellow) is visualized in Figure 2 (b) and its calculation detailed in the supplementary section A.4. We notice that this distance is the lowest for the body and legs, which define the elephant's gallop, whereas the movement of the head does not follow the periodic pattern.

For the TRUCK and YARIS datasets, the goal is the detection of clusters corresponding to different deformation patterns in the components' embeddings. This speeds up the analysis of car crash simulations since relations between model parameters and the deformation behavior are discovered more easily [32, 37]. In the 2D visualizations for the TRUCK components, we detect two clusters corresponding to a different deformation behavior and our patch-based approach allows us to identify the patches that contribute most to this. For each patch, we define a score, which equals the accuracy of an SVM (between 0.5 and 1) that is classifying the observed two deformation patterns of the entire component from the patch's embedding, see Figure 2 (c) (high similarity in yellow). The highlighted patches correlate to the left part of the beam, where the deformation is visibly different for two different TRUCK simulations in Figure 5. Note, that this comparison of patch- and shape-embeddings does not lead to significant results for the spatial CoSMA [2] because of the instability of its results.

For the YARIS, which has never been seen by the network during training, we also visualize the low-dimensional representation for different components in 2D using PCA. We detect a deformation pattern in the front beams that splits up the simulation set into two clusters, see Figure 11 in the supplementary material, which is a result similar to using a nonlinear dimensionality reduction in [2].

**Real Samples**     **Generated Samples**     **Real Samples**     **Generated Samples**

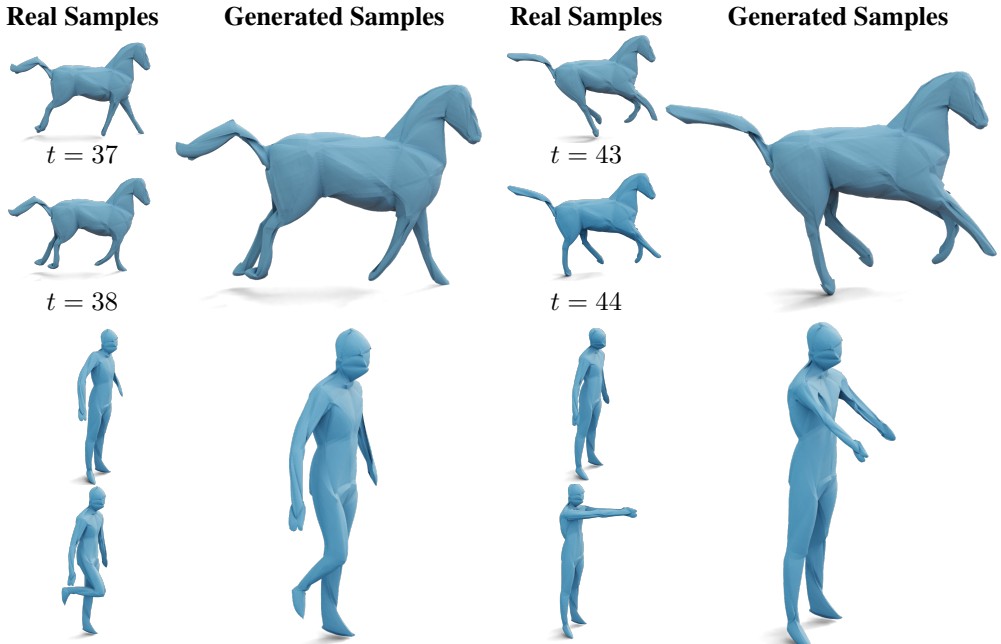

$t = 37$

$t = 38$

$t = 43$

$t = 44$

**Figure 6:** Generating new shapes by averaging the embeddings of the two shapes on the left, visualized by their reconstructions, and input these new patch-wise embeddings into the decoder only. Since the FAUST dataset contains no sequences but single shapes, the last interpolation has too short arms, since the arm-trajectory is not contained in the dataset.

## 4.5 Interpolation in the Embedding Space

To evaluate our model in a generative approach, we interpolate in the low-dimensional shape space and decode the resulting embeddings. Figure 6 shows generated samples passing averaged embeddings of two known shapes to the decoder of either the GALLOP or FAUST trained autoencoder. The generated shapes are smooth, well-formed, and resemble an average position in between the two real samples. This shows that our model is not overfitting to the training shapes.

## 5 Conclusion

We have introduced a novel spectral mesh autoencoder pipeline for the analysis of deforming 3D semi-regular surface meshes with different connectivity. This allows us to generate high-quality reconstructions of unseen meshes, that have not been presented during training. In fact, the reconstruction quality for unknown meshes with our spectral CoSMA is higher than with baseline autoencoders that have seen the meshes during training. Also, we identify and rectify artifacts due to the patch boundary handling in the surface-aware loss calculation. These improved transfer learning and generalization capabilities, the increased reconstruction quality, and the first results of using our model in a generative approach motivate the future analysis of generative models for the patch-based approach. For high-quality generative results, we also plan to improve the remeshing procedure to focus more on detailed structures. In addition, we provide an understanding and interpretation of which surface areas lead to the patterns in the embedding space. We assume that such information per patch can be used in further analysis.

An open question is, how to build mesh-independent decoders or mesh-generative models. Our mesh autoencoder can be trained for different meshes at the same time, but still requires a given mesh topology, whose vertex coordinates are reconstructed. To the best of our knowledge, it is an open question, of how to reconstruct meshes, when no template mesh is given.

## Acknowledgement

This work was developed in the Fraunhofer Cluster of Excellence »Cognitive Internet Technologies«.

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

# A  Supplementary Material

## A.1  Ablation Study and Runtime Analysis

We perform an ablation study to justify some of the design and parameter choices in our spectral CoSMA architecture. In Table 4, we report the P2S errors on the FAUST dataset and the elephant from the GALLOP dataset after 50 epochs of training. The accuracy degrades for at least one of the two datasets when we reduce the degree $K$ of the Chebyshev polynomials, reduce the size of the hidden representation $hr$, reduce the number of output channels of the convolutional layers, or change the Chebyshev Graph Convolution to the Graph Convolution from [10], who use only first-order Chebyshev polynomials. For the latter change, the networks are trained for 100 epochs.

We also list the P2S errors for training without using the surface-aware training loss but instead, the patch-wise mean squared error and a hidden representation of size $hr = 8$ as in [2]. These networks are trained for 150 epochs as the main experiments. The last line in Table 4 in comparison to the Spatial CoSMA [2] results in Table 2 show the improvement by switching from spatial to spectral convolutional layers.

In Table 5, we provide reconstruction errors when training our spectral autoencoder for semi-regular meshes for each animal separately. Notice that the reconstruction errors for horse and camel stay the same, but the reconstruction error for the elephant decreases once it is considered a training shape.

Additionally, Table 6 lists the vertex-to-vertex mean squared reconstruction errors.

**Table 4:** Ablation study of our parameter choices based on P2S errors ($\times 10^{-2}$) for 2 training runs.

| Model | P2S Error | |
|---|---|---|
| | FAUST | Elephant |
| full | **0.0031 + 0.006** | **0.0054 + 0.012** |
| $hr = 8$ | 0.0053 + 0.010 | 0.0083 + 0.016 |
| $K = 4$ | 0.0031 + 0.006 | 0.0055 + 0.012 |
| $2^3$ and $2^4$ channels | 0.0031 + 0.006 | 0.0060 + 0.013 |
| GCN [10] | 0.0032 + 0.006 | 0.0056 + 0.012 |
| Patch-wise train MSE | 0.0033 + 0.006 | 0.0074 + 0.015 |
| $hr = 8$ and patch-wise train MSE as in [2] | 0.0041 + 0.007 | 0.0085 + 0.016 |

**Table 5:** Point to surface (P2S) errors ($\times 10^{-2}$) between reconstructed unseen semi-regular meshes ($rl = 4$) and original irregular mesh and their standard deviations for three different training runs. Animals are considered as separate datasets as for the mesh-dependent baselines.

| Mesh Class | P2S Error: Our Model |
|---|---|
| Horse | 0.0022 + 0.005 |
| Camel | 0.0030 + 0.006 |
| Elephant | 0.0050 + 0.011 |

**Table 6:** Vertex-to-vertex reconstruction errors ($\times 10^{-2}$) between reconstructed and original unseen semi-regular meshes ($rl = 4$) and their standard deviations for three different training runs.
*: the elephant has not been seen by the network during training.

| Mesh Class | CoMA [16] | Neural3DMM [4] | SubdivNet [13] | Spatial CoSMA [2] | Ours |
|---|---|---|---|---|---|
| FAUST | 14.126 + 28.20 | 5.974 + 11.87 | 11.376 + 15.90 | 0.088 + 0.14 | **0.011 + 0.06** |
| Horse | 0.031 + 0.11 | 0.055 + 0.28 | 0.047 + 0.07 | 0.029 + 0.04 | **0.009 + 0.02** |
| Camel | 0.037 + 0.11 | 0.071 + 0.33 | 0.043 + 0.06 | 0.034 + 0.04 | **0.010 + 0.02** |
| Elephant | 0.041 + 0.12 | 0.075 + 0.41 | 0.060 + 0.09 | 0.106 + 0.17 * | **0.017 + 0.04** * |

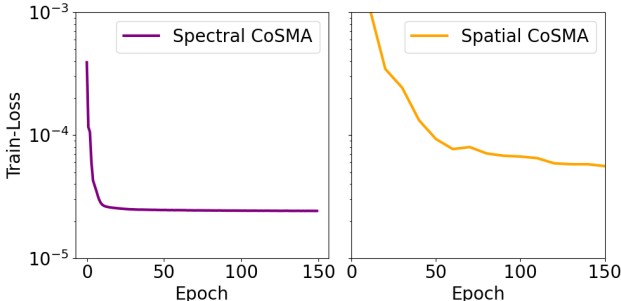

**Figure 7:** Training error (Vertex-to-vertex mean squared error measured for each patch) per Epoch for the GALLOP dataset and $rl = 4$ for the training of the CoSMA networks.

**Table 7:** Runtime of different CoSMAs per epoch when training on GALLOP and FAUST datasets using a batch size of 100.

| Mesh Class | Spatial CoSMA | | Ours | |
|---|---|---|---|---|
| | $rl$=3 | $rl$=4 | $rl$=3 | $rl$=4 |
| FAUST | 17.3 sec | 18.7 sec | 6.9 sec | 11.8 sec |
| GALLOP | 16.7 sec | 17.8 sec | 10.1 sec | 17.2 sec |

The runtime analysis of the CoSMA architecture shows, that our spectral CoSMA has a similar runtime per epoch for $rl = 4$ when comparing it to the spatial CoSMA, see Table 7 for GALLOP and FAUST datasets. For $rl = 3$ the runtime is reduced by 50% because the spectral CoSMA's runtime scales with the refinement level. For a more detailed comparison, we illustrate the validation error per epoch in Figure 7 when training both networks with the patch-wise training error. It shows, that the spectral CoSMA converges in six times fewer epochs in comparison to the spatial CoSMA. This means that the total training time of a spectral CoSMA is reduced by more than 75% for $rl = 4$. The training has been conducted on an Nvidia Tesla V100.

### A.2 Surface-Aware Loss Calculation

Given a semi-regular mesh with $n$ vertices, that is made up of $k$ patches, which have $m$ vertices without considering the padding. For all vertices, $P_i$ is the set of patches, in which vertex $i$ appears. Then, we calculate the patch-wise surface-aware training loss between the ground truth 3D coordinates $x_p$ of the patch $p$ and their reconstructions $x_p^*$ as follows:

$$\mathtt{MSE}_{SA}(x_p, x_p^*) = \frac{1}{m} \sum_{i=1}^{m} \frac{1}{|P_i|} ((x_p)_i - (x_p^*)_i)^2$$

When considering the MSE for the whole mesh, it holds

$$\frac{1}{k} \sum_{p=1}^{k} \mathtt{MSE}_{SA}(x_p, x_p^*) = \frac{1}{k} \sum_{p=1}^{k} \frac{1}{m} \sum_{i=1}^{m} \frac{1}{|P_i|} ((x_p)_i - (x_p^*)_i)^2$$

$$= \frac{1}{km} \sum_{p=1}^{k} \sum_{i=1}^{m} \frac{1}{|P_i|} ((x_p)_i - (x_p^*)_i)^2$$

$$= \frac{1}{km} \sum_{i=1}^{n} \sum_{p \in P_i} \frac{1}{|P_i|} ((x_p)_i - (x_p^*)_i)^2$$

and the reconstructions of all vertices have the same weight, taking the average if there are multiple reconstructions.

Note in Figure 8 we show how a patch-wise training without using the surface-aware loss and therefore over-weighting the patch-boundaries leads to flat patches, whose curvature is not captured

by the network. Table 4 contains the reconstruction errors when using the patch-wise train MSE in comparison to the surface-aware loss calculation during training.

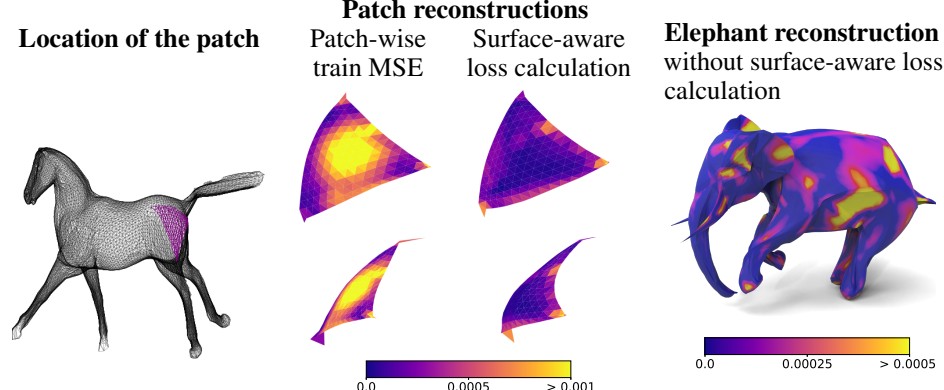

**Figure 8:** Comparison of reconstructed patches of the spectral CoSMA networks without and with using the surface-aware loss calculation during training. We highlight the face-wise reconstruction errors for the highlighted patch, which are averaged over time. Additionally, we provide the elephant's reconstruction without using the surface-aware training loss.

### A.3    Out-of-Distribution Generalization

Since the patch-wise deformations of GALLOP and FAUST are both of natural origin, we test the out-of-distribution generalization of our spectral CoSMA. We train the model on one and attempt reconstruction on the other dataset. This experiment's results are provided in Table 8. Generally, one can notice that the reconstruction errors are only slightly higher when applying the FAUST-trained network to the GALLOP testing samples. When training the network on the GALLOP dataset, the reconstructions on the FAUST test samples are as good as when trained on the dataset. This seems surprising and might be due to the increased size and variability of the patches in the GALLOP training dataset. Also, the patch-wise approach is convenient since it focuses on the local patch deformation, which is of natural origin for both datasets.

**Table 8:** Point to surface (P2S) errors ($\times 10^{-2}$) between reconstructed and original unseen semi-regular meshes ($rl = 4$) and their standard deviations for three different training runs. The model is tested and trained on different datasets.

| Testing Dataset | Training Dataset | P2S Errors: Our Model |
|---|---|---|
| FAUST | GALLOP | 0.0030 + 0.005 |
| Horse | FAUST | 0.0022 + 0.005 |
| Camel | FAUST | 0.0033 + 0.006 |
| Elephant | FAUST | 0.0055 + 0.012 |

### A.4    Additional Reconstructed Samples and 2D Visualizations of the Embeddings

We provide additional reconstructed samples from the GALLOP and FAUST datasets in Figure 9. Additionally, Figure 10 compares reconstructed patches from the two CoSMA approaches. It is visible that the reconstruction from the novel spectral CoSMA is smoother.

Figure 11 shows the embeddings in the low-dimensional space for two YARIS front beams. The beams deform in two different branches, which manifests in the embedding.

For the GALLOP dataset, we calculate the distance between the patch-wise embeddings and the embedding of the entire shape, to determine how important the patch's deformation is for the general deformation behavior of the whole shape. We interpolate and densely subsample the lines connecting the embedding points of consecutive timesteps. Between the sampled points $p_i^s$ describing the

deformation of the entire shape over time and the sampled points $p_j^p$ from the patch's embedding, we calculate a chamfer distance, since the embedding shape is cyclic. The chamfer distance [38] measures the average squared distance between each point $p_i^s$ to its nearest neighbor from all points $p_j^p$ and vice versa. Therefore the distance is the lowest for circle-like patch-wise embeddings.

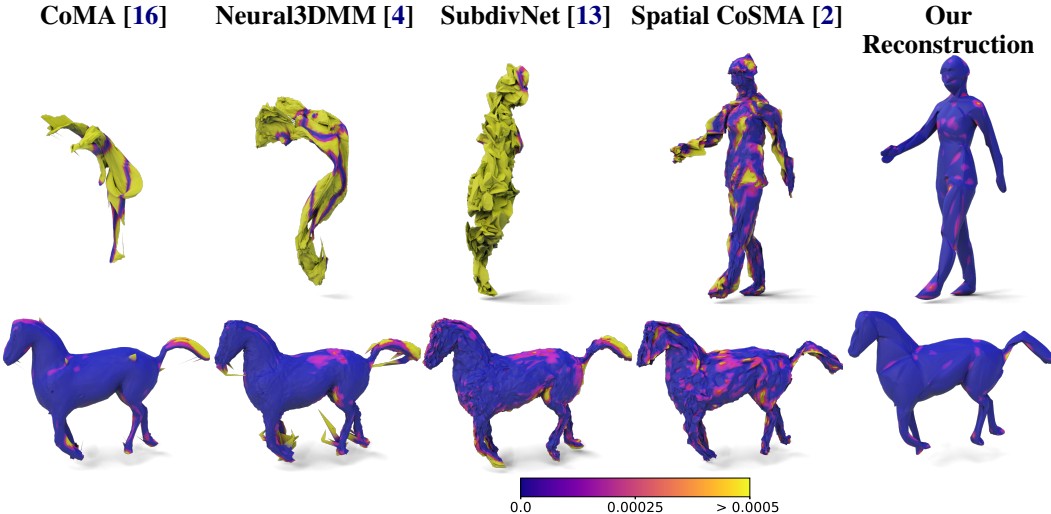

**Figure 9:** Additional reconstructed unknown FAUST pose and reconstructed horse test sample at $t = 39$ by CoMA, Neural3DMM, SubdivNet Autoencoder, spatial CoSMA, and our network with highlighted P2S error.

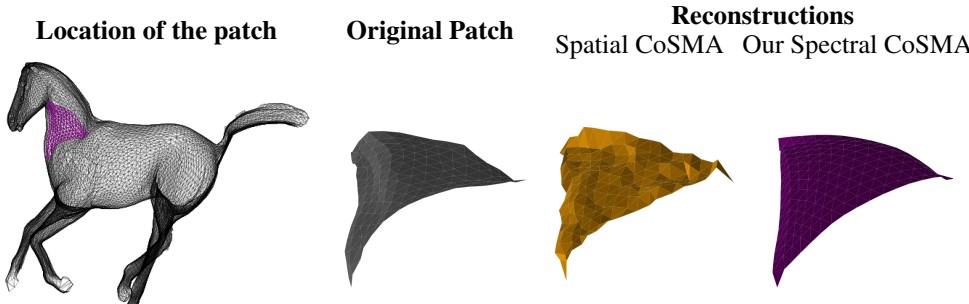

**Figure 10:** Comparison of reconstructed patches of the CoSMA networks.

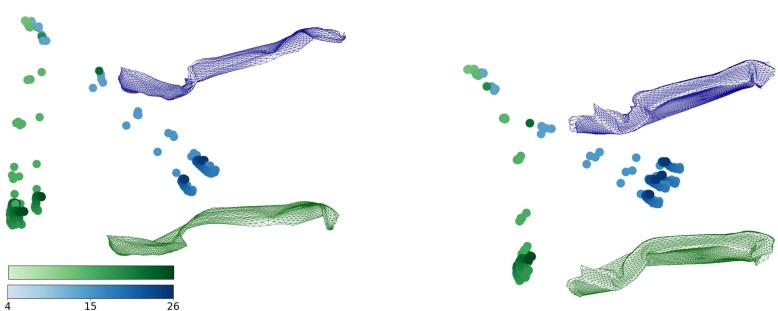

**Figure 11:** Spectral CoSMA embeddings of the YARIS front beams for 10 simulations, which deform in two branches. Color encodes timestep and branch.

## A.5 Model Parameters and Reconstruction Errors for Refinement Level 3

For the baselines and our spectral CoSMA, we list the number of trainable parameters of the models for the different meshes in refinement level $rl = 3$ and $rl = 4$. Increasing the refinement level by one, increases the number of faces by a factor of four.

**Table 9:** Number of vertices per mesh and trainable parameters for the reconstruction of semi-regular meshes using refinement level 4.

| Mesh Class | # Vertices irregular | # Vertices semi-regular | CoMA [16] | Neural 3DMM [4] | SubdivNet [13] | Spatial CoSMA [2] | Ours |
|---|---|---|---|---|---|---|---|
| FAUST | 6890 | 12,772 | 46,379 | 426,195 | 879,857 | 26,888 | 23,053 |
| Horse | 8,431 | 14,745 | 50,731 | 459,987 | 1,010,417 | | |
| Camel | 21,887 | 12,802 | 46,923 | 430,419 | 879,857 | 26,888 | 23,053 |
| Elephant | 42,321 | 15,362 | 52,363 | 472,659 | 1,053,937 | | |

**Table 10:** Comparison of the number of parameters for meshes of refinement level 3 from [2].

| Mesh Class | CoMA [16] | Neural 3DMM [4] | Spatial CoSMA [2] | Ours |
|---|---|---|---|---|
| FAUST | 26,795 | 276,275 | 18,184 | 16,235 |
| Horse | 27,339 | 280,499 | | |
| Camel | 26,795 | 292,659 | 18,184 | 16,235 |
| Elephant | 27,339 | 296,883 | | |

**Table 11:** Point to surface (P2S) errors ($\times 10^{-2}$) between reconstructed unseen semi-regular meshes ($rl = 3$) and original irregular mesh and their standard deviations for three different training runs. Additionally, the average Euclidean vertex-wise error (in cm) is given.
$^*$: the entire YARIS dataset has not been seen by the network during training.

| Dataset | Component Lengths | Spatial CoSMA [2] | | Ours | |
|---|---|---|---|---|---|
| | | Test P2S | Eucl. E. | Test P2S | Eucl. E. |
| TRUCK | 135–370 cm | 0.0443 + 0.071 | 2.23 cm | 0.0043 + 0.009 | 0.43 cm |
| YARIS$^*$ | 21–91 cm | 0.1784 + 0.380 | 0.80 cm | 0.0458 + 0.090 | 0.37 cm |

## A.6 SubdivNet Autoencoder Architecture

We translated our spectral CoSMA architecture to the SubdivNet baseline by replacing the Chebyshev Convolutions with the Subdivision-Based Mesh Convolutions and the corresponding pooling and unpooling operators introduced in [13], see Table 12. All SubdivNet convolutions use stride and dilation equal to one, kernel size equal to three, and are followed by ReLU activations. As the SubdivNet convolutions operate on face features instead of vertex features, we used the coordinates of the three adjacent vertices per face as input features. The bullets • reference the corresponding batch size. The data's second dimension is the number of features and the last dimension is the number of faces of the current mesh.

**Table 12:** Structure of the autoencoder used for the SubdivNet Baseline.

| Encoder Layer | Output Shape | Param. | Decoder Layer | Output Shape | Param. |
|---|---|---|---|---|---|
| Input | $(\bullet,\ 9, 25600)$ | 0 | Fully Connected | $(\bullet, 2^5,\ 1600)$ | 460800 |
| MeshConv | $(\bullet, 2^4, 25600)$ | 592 | MeshUnpool | $(\bullet, 2^5,\ 6400)$ | 0 |
| MeshPool | $(\bullet, 2^4,\ 6400)$ | 0 | MeshConv | $(\bullet, 2^5,\ 6400)$ | 4128 |
| MeshConv | $(\bullet, 2^5,\ 6400)$ | 2080 | MeshUnpool | $(\bullet, 2^5, 25600)$ | 0 |
| MeshPool | $(\bullet, 2^5,\ 1600)$ | 0 | MeshConv | $(\bullet, 2^4, 25600)$ | 2064 |
| Fully Connected | $(\bullet, 8)$ | 409608 | MeshConv | $(\bullet,\ 9, 25600)$ | 585 |

