# OpenReview forum: "Transfer Learning using Spectral Convolutional Autoencoders on Semi-Regular Surface Meshes"
_logconference.io/LOG/2022/Conference — LoG 2022 Poster_

### Official Review · Reviewer_ytYM · 2022-09-22

**Overall Score:** 5
**Confidence:** 4

**Review:**

The paper introduces a new autoencoder architecture for deformable 3D meshes. The proposed pipeline is a spectral variant of an existing method (convolutional semi-regular mesh autoencoder, CoSMA for short), where the authors demonstrate that shifting from a spatial to a spectral paradigm brings several advantages, including better reconstruction quality / fidelity, less learnable parameters, and robustness to different meshing. These points are demonstrated on three different datasets in comparison with other 3D autoencoders.

From a general perspective, the paper presents some new ideas that are well executed, but at the same time it suffers from a limited experimental evaluation and unclear positioning with respect to existing approaches. More in detail:

(+) The paper is well written for the most part, and is easy to follow for a reader with some background knowledge on geometry processing and deep learning. The provided illustrations are useful and well made; for example, I found Figure 5 especially illuminating -- I suggest to move it earlier in the paper, perhaps in the methodological section rather than in the last part of the experiments.

(+) The idea of applying spectral convolutions to regular patches is straightforward yet effective, and novel to my knowledge.

(+) Some of the experimental results look impressive, most notably the reconstruction quality for the elephant class, despite the latter having never been seen at training time.

(-) One important aspect that I feel the paper should address, is a clearer positioning with respect to prior art. In particular, throughout the paper it is not always clear what consitutes a methodological novelty of the paper, and what is directly inherited from spatial CoSMA and other building blocks in the pipeline. For example, the spectral approach is described as being mesh-independent (e.g. line 52 in the introduction) due to the adoption of regular patches that replace the original mesh representation; however, to my understanding this is not a key feature that is unique to the proposed spectral variant, but rather it holds for any method that adopts the semi-regular mesh representation of surfaces, such as spatial CoSMA itself and SubdivNet.

(-) Does the training procedure require shapes in correspondence (i.e. full point-to-point maps)? Please comment on this.

(-) Figure 3 shows qualitative comparisons with other approaches, but the actual improvement is hard to judge due to the fact that three out of four compared methods use a different training set (e.g. where the elephant class is included). It would be better to show full comparisons for both cases, that is with and without the additional class in the training set.

(-) The experimental evaluation lacks what I believe are important experiments for a paper introducing a new autoencoder architecture. These include: (i) showing the qualitative performance of the autoencoder as a generative model, by randomly sampling the constructed latent space and decoding the random samples to synthesize new shapes; this is useful to demonstrate the lack of overfitting and the capability of the model to correctly encode a shape space with well-formed shapes; (ii) decoding interpolations between latent codes, along the lines of the previous experiment, but useful to show the kind of deformations the AE is able to encode as opposed to single shapes taken individually; the example in Figure 5a goes in this direction, but is shown on a path that was completely seen at training time, while one is also interested in generating completely new paths and therefore new animations; (iii) importantly, to date the best AE models for 3d shapes, at least in terms of reconstruction quality, are those based on signed distance functions (SDF) and other implicit representations (see the line of works of Lipman et al, among others); for this reason, it would be important to at least mention them and discuss in what settings is a mesh-based AE preferrable. Otherwise, there is a risk the proposed method will be overlooked in favor of higher fidelity SDF-based approaches.

Some minor comments follow:

* The paper uses the term "transfer learning" to refer to out-of-distribution generalization and robustness to meshing. I suggest to avoid using this terminology to avoid confusion, since some readers might interpret it in terms of cross-network weight transfer, or knowledge distillation and related methods.

* Line 95 mentions CoMA (2018) being the first convolutional mesh autoencoder, but I believe others precede it, see e.g. "Deformable Shape Completion with Graph Convolutional Autoencoders", Litany et al 2017.

* Lines 125-127 mention that the remeshing procedure does not change the underlying continuous surface, but this is not true in general. A more conservative statement might be preferable, along the lines of "remeshing the polygonal mesh only induces a small, bounded distortion on the underlying surface, in some cases maintaining it exactly the same".

* I wonder if the described remeshing procedure is necessary at all: since the aim is to produce a patch with regular meshing, why not just start from a canonical template patch and then fit it to the data using e.g. least-squares meshes [Sorkine and Cohen-Or 2004]?

* Is the data augmentation of line 251 applied to each patch individually (each patch has its own rotation), or coherently to all patches?

To conclude, the idea of shifting from a spatial to a spectral patch-based convolution is meaningful and might possibly lead to a good impact, but the experimental evaluation requires some additional work to deliver a complete and solid picture. The adoption of a spectral convolution is not the only contribution, however, throughout the paper it is not easy to trace the line between what is actually novel and what is inherited from existing approaches. For these reasons, at this stage I believe the paper would require a major revision to be considered for publication.

---

### Official Review · Reviewer_zDJe · 2022-10-10

**Overall Score:** 8
**Confidence:** 4

**Review:**

Description:
The paper proposes a spectral method for 3D reconstructions of semi-regular surface meshes. The entire irregular mesh is “remeshed” into coarser meshes/patches, which is then sub-divided into regular meshes to obtain the semi-regular mesh. The spectra autoencoder is applied to the patches. For the encoder,  a series of spectral convolutions(Chebysev) is then applied to each of the patches followed by pooling to obtain coarser representations. Finally an MLP is used to get the hidden representation. For decoding, the encoder is reversed with unpooling layers. The pooling/unpooling layers are designed such that the 4 faces in a 2-simplex are  united/divided into one/four with the embeddings averaged/interpolated. The authors show that their method is able to generalize to unseen shapes and have lower reconstruction errors than baselines.

Strengths:
1) The reconstruction error is lower compared to baselines
2) Better generalization/learning to new shapes not seen during training
3) Lower parameters than baselines that model the entire mesh
4) Ability to model the global deformation in the patch as compared to the spatial patch baseline(spatial CoSMA)

Weaknesses:
1) Novelty is limited as the spatial CoSMA has been adapted to spectral using ChebNets instead of spatial convolution

Comments/Limitation/Concerns:
1) Since the authors claim the proposed “surface aware” loss to be beneficial to the task, the paper should detail some equations on it.
2) While the paper claims to achieve transfer learning, it is only shown within a dataset between different meshes/objects. Would it also work(give low reconstruction error) on different datasets i.e. train on one and attempt reconstruction on another?
3) Regarding baselines, it may not be fair to compare directly with the complete mesh baselines as they solve a harder problem of reconstructing from the complete mesh. This may be a limitation of the current method. It would be interesting to see if the proposed method works in that setting too.
4) While the authors mention the limitation with the MSE metric used in the baseline(spatial CoSMA), could the MSE also be reported along with P2S. This would give more perspective into the performance of the proposed method with respect to the baseline.

Overall the problem is an important one and the use of spectral techniques to enhance the reconstruction seems sound. However, this paper only performs patchwise reconstructions and in this sense most of the baselines are not fair to compare with. It is not clear how to use the technique in practice for whole object/mesh reconstruction. Also certain technical details could be better explained. Thus I originally voted for a weak accept.

In the rebuttal the authors provide additional results that demonstrate ability to transfer the learning over different datasets, show the effect of proposed surface aware loss etc. among other results. The discussion also helped clarify the potential to use the method for whole mesh generation and broader applicability. The revised manuscript alleviates my concerns and so I change my score to a clear accept.

---

### Official Review · Reviewer_JYtR · 2022-10-20

**Overall Score:** 3
**Confidence:** 3

**Review:**

A transfer learning with a spectral graph convolutional filter is presented on semi-regular meshes in this article.

Strong points include the following:
(1) the results show a significant improvement beyond baselines;
- in terms of error, the newly proposed method decreases the error to one less magnitude, as shown in Tables 1 and 2;
- the visualization of figs. 3 and 4 demonstrated the improvement in actual cases;
- the proposed method decreases the error to one less magnitude than was previously possible.
(2) The implementation is simple to comprehend
- as it is based on numerous commercially available tools; the implementation description can be found in section 4, and it has an accurate presentation of all the relevant elements.

Points of weakness
(1) The proposal has a limited originality because it is an incremental improvement based on previous efforts.
- this work is heavily dependent on the work of related researchers [23]
- Since many of the applied tools have been around for a while, this work is simply a combination of those older applications, such as spectral filtering and padding, among others.
(2) The presentation of the model description is not obvious;
- even while the details are offered, it is not clear what the most important novelty is, nor is it clear what the actual network structure is. It would be helpful if the author could include a visualization of the structure of the network.
- In addition, it is not entirely clear why using spectrum analysis is preferable. It's possible that the author will provide some kind of theoretical analysis or demonstration. *Levie, Ron, et al. "Transferability of Spectral Graph Convolutional Neural Networks." J. Mach. Learn. Res. 22 (2021): 272-1." is a paper that was published recently and discusses the transferability of spectral filters.
- The third section acts as a preliminary introduction, but it does not go into detail. In that case, it can be combined with the second section.

My inquiries include the following: what, beyond the scope of the linked work, is their most significant contribution, particularly [23]?

---

### Official Review · Reviewer_Mtun · 2022-10-21

**Overall Score:** 6
**Confidence:** 4

**Review:**

Summary: This work improves the work of Hahner and Garcke (reference 2 in the paper), a.k.a. Spatial CoSMA, by using spectral rather than spatial convolutions of the semiregular patches. This architectural choice is experimentally shown to significantly improve the transfer learning performance of the method, in terms of reconstruction quality of the autoencoder, outperforming other competing mesh autoencoder methods. The authors show the model’s ability to analyze surface deformation patterns from low dimensional representations.

Pros:

-Through experimental results and ablation studies, the authors show convincing benefits for the spectral based convolution rather than spatial convolution on mesh reconstruction

-Usefulness of learned embeddings is also clearly demonstrated through experimental results

The point-to-surface evaluation metric is good for comparison with the ground truth mesh

Cons:

-The datasets used, such as GALLOP and FAUST, have a lot of meshes that share the exact same connectivity within each group (e.g. each animal in GALLOP has the same connectivity). The low dimensional embeddings for each shape was generated by concatenating the embeddings of each individual patch. This seems to work only if you have the same number of faces, and same ordering of faces in the meshes, which is the case for those datasets. This seems to be limiting on the types of meshes one could use this method for latent space examination of the whole shape.

Further Comments/Questions:

-It would be good to provide details on how the SubdivNet-based autoencoder was implemented, since that paper does not explicitly construct an autoencoder for reconstruction of meshes.

-It might be useful to show if the method is robust to other remeshing methods mentioned (e.g. MAPS) when applicable.

-For the baseline methods that don’t use remeshing, are they trained on the remeshed meshes, or the original meshes?

-Continuing on the previous question, it may also be helpful to report the MSE with the remeshed ground truth meshes for any methods that are trained on the remeshed meshes.

---

### Meta-Review · Area_Chair_253V · 2022-11-15

**Confidence:** 4
**Recommendation:** Accept

**Meta Review:**

Final reviewers scores are "Clear Accept", 2 "Weak Accept" and "Weak Reject".

## Strengths
(S1) Reviewers appreciated that while single parts of the proposed method aren't novel, the way they are put together is.
(S2) In paper and rebuttal together, the authors have provided experiments that most reviewers feel provide sufficient evidence for the claims.

## Weaknesses
(W1) Reviewers were confused about the novelty of this method over prior work, though some concerns were alleviated in the rebuttal.
(W2) One reviewer was confused about the literal network architecture and inner workings of the method.

I thus recommend Acceptance of this manuscript, with the following weighting of importance: (S2) > (S1) > (W2)

---

### Decision · Program_Chairs · 2022-11-22

Accept (Poster)